# Fishing Participation, Motivators and Barriers among UK Anglers with Disabilities: Opportunities and Implications for Green Social Prescribing

**DOI:** 10.3390/ijerph19084730

**Published:** 2022-04-14

**Authors:** Rosie K. Lindsay, Christina Carmichael, Peter M. Allen, Matt Fossey, Lauren Godier-McBard, Laurie Butler, Mike Trott, Shahina Pardhan, Mark A. Tully, Jason J. Wilson, Andy Torrance, Lee Smith

**Affiliations:** 1Vision and Hearing Sciences Research Centre, Anglia Ruskin University, Cambridge CB1 1PT, UK; peter.allen@aru.ac.uk; 2Centre for Health, Performance and Wellbeing, Anglia Ruskin University, Cambridge CB1 1PT, UK; christina.carmichael@aru.ac.uk (C.C.); lee.smith@aru.ac.uk (L.S.); 3Veterans and Families Institute for Military Social Research, Anglia Ruskin University, Chelmsford CMI 1SQ, UK; matt.fossey@aru.ac.uk (M.F.); lauren.godier-mcbard@aru.ac.uk (L.G.-M.); 4The Centre for Mental Health, London W1G 0AN, UK; 5Faculty of Science and Engineering, Anglia Ruskin University, Cambridge CB1 1PT, UK; laurie.butler@aru.ac.uk; 6Vision and Eye Research Institute, School of Medicine, Anglia Ruskin University, Cambridge CB1 1PT, UK; mike.trott@aru.ac.uk (M.T.); shahina.pardhan@aru.ac.uk (S.P.); 7School of Medicine, Ulster University, Londonderry BT48 7JL, UK; m.tully@ulster.ac.uk; 8Sport and Exercise Sciences Research Institute, School of Sport, Ulster University, Newtownabbey BT37 0QB, UK; jj.wilson@ulster.ac.uk; 9Angling Direct PLC, Norfolk NR13 6LH, UK; andy.torrance@anglingdirect.co.uk

**Keywords:** angling, blue prescribing, wellbeing, health

## Abstract

Green social prescribing, which includes the referral of patients to nature-based activities, could exacerbate inequalities between people with disabilities and people without. Research suggests fishing could be more inclusive relative to other outdoor sports. To understand if fishing is an inclusive sport, and the potential benefits and barriers to prescribing fishing, the present study compared participation, motivators and barriers to fishing, between anglers with and without disabilities. UK adults were invited to participate in an online survey. Chi-square tests examined differences between anglers with and without disabilities regarding the type of fishing anglers engaged in, the frequency of fishing, the length of time spent fishing, motivators for fishing and barriers to fishing. Among 1799 anglers (97.5% male), 292 (16.2%) anglers reported having a disability. Most anglers with disabilities were over 55 years old (56.5%). There was no difference in fishing participation, or motivators for fishing, between anglers with and without disabilities; however, anglers with disabilities were more likely to report ‘costs’, ‘lack of transport’ and ‘having no one to go with them’ as barriers. Overall, there appeared to be no differences in fishing participation between anglers with versus without disabilities, although additional barriers to participation may exist.

## 1. Introduction

One in five people in the UK have a disability [1]. The Equality Act 2010 defines disability as ‘a physical or mental impairment that has a ‘substantial’ and ‘long-term’ (12 months or more) negative effect on your ability to do normal daily activities’ [2]. In 2021, the Office for National Statistics reported that anxiety ratings were twice as high for people with a disability, and people with a disability were four times more likely to report feeling lonely ‘often’ or ‘always’ when compared to people without a disability [3]. People with a disability also self-reported poorer ratings of happiness, feeling like things done in life are worthwhile and life satisfaction. From 2014 to 2021, there have been no improvements in this disparity [3,4]. Further to this, people with disabilities are more likely to have unmet healthcare needs due to barriers to healthcare such as cost, long waiting lists and transportation problems [5]. Thus, the problem is cyclical. People with disabilities are more likely to have poorer physical and mental health, and are less likely to receive the healthcare support they need.

Disparities in health between people with disabilities and people without are further compounded by inequities in access to health and wellbeing services within the community. The UK Office of National Statistics reported that 27.6% of people with disabilities participated in “sports or exercise” groups compared to 43.1% of people without disabilities in 2018 [6]. Concerningly, an Activity Alliance survey found 51% of people with a disability agreed that physical activity was for someone like them, compared to 77% of people without a disability [7]. Lower participation in organised sport and exercise is likely to contribute to lower physical activity levels among populations with a disability. Research has found that people with a disability are twice as likely to be inactive as people without [8]. Another contributor to lower physical activity among populations with disabilities is poorer access to nature. Data collected over six waves of survey data (2009/2010–2015/2016) from over 60,000 adults in England found that adults with disabilities were 95% less likely to visit nature at least once a month, compared to adults without disabilities [9]. Inequities in access to nature and physical activity excludes people with disabilities from the range of benefits outdoor physical activity can offer. A 2019 review of 133 studies found evidence that outdoor sport provides benefits to physical health, mental health, wellbeing and social connectedness, as well as wider benefits such as education and lifelong learning, active citizenship, crime reduction and reduced anti-social behaviour [10].

Inequities in access to physical activity, as well as inequities in outdoor space, is concerning given that the UK National Health Service (NHS) is expanding green social prescribing initiatives, as part of the plan to refer at least 900,000 people to social prescribing by 2023/24 [11]. Green social prescribing refers to the process of healthcare professionals referring patients to “nature-based interventions and activities, such as local walking for health schemes, community gardening and food-growing projects” [12]. Green social prescribing initiatives refer patients to both green and blue space [13] (natural aquatic environments) [14]. However, many of the green social prescribing initiatives involve outdoor groups and services that require people to take part in higher intensities of physical activity to fully take part. These risks exacerbating existing inequalities in health and access to healthcare for those who have disabilities which limit their ability to engage in high levels of moderate-to-vigorous activity. Thus, there is a need to improve access, opportunities and inclusivity within green social prescribing initiatives. However, it is also important to identify existing opportunities which are currently inclusive, and could be included as part of green social prescribing to target people with disabilities.

One outdoor sport which may be more inclusive for populations with disabilities relative to other outdoor sports is recreational fishing (referred to as fishing here on). Research suggests that fishing can be beneficial for increasing physical activity, for example walking to the venue, wading into the river to fish and casting the rod are all forms of activity associated with fishing [15]. Importantly, fishing is generally classified as light-to-moderate physical activity [16], therefore, fishing may be a more accessible form of outdoor physical activity for populations with conditions that limit their ability to engage in vigorous physical activity. Previous studies have found even small increases in physical activity significantly reduce the risk of all-cause mortality, when compared to doing no physical activity [17]. In this sense, fishing could be an important tool for engaging people with disabilities who are at high risk of being physically inactive, as they are likely to gain large benefits from only small increases in their physical activity levels. As fishing is an outdoor sport, it could also provide the benefits associated with being in nature as previously discussed. Fishing also makes use of outdoor water environments and research has shown that the use of blue space in interventions can benefit health, particularly mental health and psycho-social wellbeing [18].

The Angling Trust resource hub proposes that fishing is an inclusive activity which can be used to relieve long term health and wellbeing conditions as part of social prescribing initiatives [19]. Previous research has found fishing can be a form of stress relief [20,21,22], improve social relations [23,24] and is an accessible form of physical activity for older adults and people recovering from illness [15]. However, it is important to examine if there are significant differences in fishing participation patterns (i.e., frequency, duration, type of fishing, and fishing match engagement), between people with disabilities and people without disabilities. Comparing participation patterns is important to understand which types of fishing are popular and if particular types of fishing are more inclusive than others. In addition, comparing duration and frequency of fishing is important as research suggests that at least 120 min a week in nature exposure is needed for significant improvements in health and wellbeing [25]. Furthermore, it is important to identify any significant differences in the motivators and barriers to fishing between people with disabilities and people without disabilities. Identifying motivators could highlight the benefits of fishing to individuals. These motivators could then be promoted to encourage more people with disabilities to fish. Identifying existing barriers would mean these could be strategically targeted to improve accessibility to fishing for people with disabilities. Exploring if there are significant differences between motivators and barriers to fishing for people with a disability, compared to people without a disability, is important for identifying differences in fishing experiences, and informing future interventions which tackle health inequalities. Therefore, the aim of this study is to compare the participation patterns, motivators and barriers to fishing between anglers with disabilities and anglers without disabilities.

## 2. Methods

A cross-sectional study design was utilised. Participants were recruited from October 2021 to January 2022 via an online survey that was advertised through Angling Direct and Tackling Minds Instagram, Facebook and Twitter accounts. Angling Direct also sent the survey link to their mailing list and the link was distributed via the Anglia Ruskin University Twitter account. The survey was open to all UK residents aged 18 and over. Only responses from those who participated in some form of recreational fishing were included in this paper. Participants provided informed consent prior to the completion of the survey and ethical approval was granted by the Anglia Ruskin University Sport and Exercise Sciences Research Ethics Board.

### 2.1. Measures

#### 2.1.1. Demographic Variables

Participants were asked to provide demographic variables including age (years), gender (Male/Female/Non-binary/Intersex/other), ethnicity (White/Black/Asian/Mixed or Multiple/Other). Participants were considered disabled if they reported ‘yes’ to ‘do you consider yourself to have a disability?’ (See Appendix A for a full list of survey questions).

#### 2.1.2. Sedentary Behaviours (Continuous Variables)

Screen time was measured by asking participants ‘on an average day, how long do you spend looking at screens? (For example, computer, tablet or phone)’. Sitting time was measured by asking participants ‘on the average day, how much time do you spend sitting?’ Participants were asked to provide their answers in hours and minutes.

#### 2.1.3. Physical Activity (Continuous Variable)

Moderate physical activity was measured by asking participants ‘on an average day, how much time do you spend doing moderate physical activity (defined as activities that take moderate physical effort and make you breathe somewhat harder than normal?’ Vigorous physical activity was measured by asking participants ‘On an average day, how much time do you spend doing vigorous physical activity (defined as activities that take hard physical effort and make you breathe much harder than normal)’. Participants were asked to provide their answers in hours and minutes.

#### 2.1.4. Fishing Participation

Participants were asked what type of fishing they engaged in. Options included: ‘coarse fishing’, ‘sea fishing’, ‘match fishing’, ‘fly fishing’, ‘carp fishing’, ‘predator fishing’, ‘specialist fishing’ and ‘other’.

#### 2.1.5. Barriers and Motivators

Participants were asked ‘what are the main barriers that might stop you fishing?’ Options included ‘weather’, ‘cost’, ‘lack of transport’, ‘lack of fishing venues nearby’, ‘no one to go with’, ‘time restrictions’ and ‘other’. Participants were able to select multiple responses. To identify motivators to fishing participants were asked ‘what is your main reason for going fishing?’. Options included: ‘to socialise’, ‘I enjoy the challenge of fishing’, ‘to be outside’, ‘to relax’, ‘to catch food to eat’ and ‘other’. Participants who reported ‘other’ barrier or motivator to fishing were asked to provide further details in a free text box.

### 2.2. Statistical Analysis

All statistical analyses were conducted using SPSS Version 28. 0.0.0 (190) (Chicago, IL, USA). To identify hours per day spent engaged in moderate-to-vigorous physical activity, the sum of time spent in moderate and time spent in vigorous physical activity was calculated. Moderate and vigorous physical activity were not examined separately as 853 participants did not assign a value to time spent engaged in moderate physical activity per day; however, all participants reported a value daily time engaged in moderate or vigorous physical activity. Chi-square tests were conducted to examine if there were differences in the type of fishing anglers engaged in, the frequency of fishing, the length of time spent fishing, motivators for fishing and barriers to fishing, which were associated with disability status (i.e., no disability vs. disability).

## 3. Results

Demographic information about the participants is shown in Table 1. A total of 1799 anglers completed the survey, 292 (16.2%) anglers reported having a disability and 1499 (83.3%) anglers reported not having a disability. Eight anglers who did not report their disability status were excluded from the analysis.

Data comparing the fishing behaviours of people with disabilities and people without are presented in Table 2. There were no statistically significant differences between disability status and type of fishing the individual participated in, (χ^2^(7) = 10.441, *p* = 0.17) frequency of fishing participation, (χ^2^(8) = 12.642, *p* = 0.13) length of time spent fishing, (χ^2^(3) = 6.030, *p*= 0.11) and participation in fishing matches (χ^2^(1) = 0.772, *p* = 0.38).

Data comparing the motivators for fishing between people with disabilities and people without are presented in Table 3. The most common motivator for fishing reported by people with disabilities and people without disabilities was that they ‘enjoy the challenge of fishing’. The second most common reason for fishing reported by people with disabilities and people without disabilities was ‘to relax’ followed by ‘to be outside’. There were no significant differences in the main motivator for fishing reported by people with disabilities compared to people without disabilities.

Data comparing the barriers to fishing between people with disabilities and people without are presented in Table 4. Most people with and without disabilities reported that the weather was a barrier to fishing. Following ‘weather’ the most common barrier to fishing reported by people with disabilities was ‘cost’, whereas for people without disabilities the second most commonly reported barrier to fishing was ‘time restrictions’. Overall, people with disabilities were more likely to report that cost (χ^2^(1) = 43.063, *p* < 0.001), lack of transport (χ^2^(1) = 21.756, *p* < 0.001), having no one to go with (χ^2^(1) = 34.144, *p* < 0.001) and ‘other’ barriers (χ^2^(1) = 27.232, *p* < 0.001) were barriers to fishing than people without disabilities. For ‘other’ most people with disabilities reported barriers related to their mental and physical health prevented them going fishing. Additional barriers people wrote for ‘other’ included inaccessible facilities, having to rely on someone to drive them to the venue, too crowded to fish, family commitments and COVID-19 restrictions at the venue.

## 4. Discussion

The present study aimed to compare the participation patterns, motivators and barriers to fishing between anglers with and without disabilities, to identify potential opportunities, motivators and barriers to offering fishing on prescription. Our results suggest that fishing participation is similar between anglers with disabilities and anglers without disabilities. The main motivators for fishing were the challenge of fishing and to relax, with no difference in motivators for fishing being shown among anglers with versus anglers without disabilities. In terms of barriers, costs, lack of transport, no one to go with and other types of barriers were more commonly reported among populations with disabilities.

The results found no difference regarding the type of fishing, frequency of fishing, length of time spent fishing and engagement in fishing matches associated with the presence of a disability. Previous studies have also suggested that fishing participation rates between people with disabilities and people without disabilities may be similar. For example, in May 2020/21, the Sport England Active Lives survey found 0.2% of people without a disability in England had engaged in angling at least twice in the last 28 days, compared to 0.3% of people with a disability [26]. The 2009/10 Sport England survey also found among 1469 anglers, 39% had a long-standing illness or disability, compared to 19% of participants in other sports [27]. To our knowledge, the current study is the first to explore and compare fishing participation behaviours in detail. With similar fishing participation behaviours being measured between both groups, this provides promising evidence suggesting that fishing is an inclusive form of sport for people with disabilities relative to other sports.

The overall demographic of anglers found in the present study is also important for future research and interventions. In line with previous research, most anglers were men [28]. Interventions which can improve mental health whilst also encouraging uptake among men are important, as approximately one in eight men in England have a common mental health problem; however, only 36% of referrals to NHS talking therapies are for men [29]. In addition, fishing participation was higher among those who were over 55 years of age. Previous studies have found that differences in physical activity between people without a disability and with a disability tend to be more pronounced in older age groups. For example, the Sport England Active lives survey 2016/17 found 53% of adults with disabilities over the age of 55 were inactive, compared to 30% of adults with disabilities aged 16–54 [30]. In summary, the results suggest fishing on prescription could be a health and wellbeing initiative which appeals to male populations, and to older populations who may currently not engage in health and wellbeing services. However, future research should explore the barriers to fishing for female populations, and younger populations with disabilities. Future research should also aim to identify other outdoor sports which may be more appealing for female and younger populations, which should be included in green social prescribing initiatives.

Our study also found that the ‘challenge of fishing’ and ‘to relax’ were the main motivators for fishing. The results highlight the potential benefits of fishing as reported by the participants. It is plausible that anglers were motivated by the challenge of fishing as overcoming challenges in sport is a mechanism to improve self-esteem [31,32,33]. Relaxation was also an important motivator for anglers, and may provide additional physical and mental health benefits, for example relaxation is important for reducing chronic stress which is associated with physical and psychiatric illnesses such as diabetes, depression and schizophrenia [34]. These findings may help healthcare professionals who are using fishing as a form of green prescription to communicate the benefits of fishing to patients. In addition, healthcare professionals may consider discussing fishing as an option with a patient who would benefit from new challenges, or relaxation.

It is also important to highlight that the barriers for people with disabilities were significantly different to people without disabilities. Cost, lack of transport and no one to go with were barriers to fishing, and poor mental and physical health were more commonly reported among populations with disabilities. Social prescribing is a patient-led process, therefore if a patient is not interested or willing to do fishing due to personal preference, it may not be appropriate to prescribe them fishing and other options should be explored. However, for people who would want to take up a prescription of fishing if no barriers to fishing were present, healthcare professionals should be aware of factors which may reduce the impact of these barriers where possible. For example, fishing-specific buddy schemes [35], social media groups and apps such as the Fishbuddy app [36] may help anglers connect with others. In addition, people with a disability may be eligible for a discounted 12-month disabled rod licence which may help to reduce some of the costs [37], and local community schemes may offer discounted travel for people with disabilities. However, environmental changes are also required to make fishing more accessible, for example providing accessible links from public transport to fisheries and ensuring accessible fishing platforms and fishing areas are provided [38].

The present study offers a unique exploration of the participation patterns of anglers with disabilities in the UK, and the motivators and barriers to fishing for anglers with disabilities, which can be used to inform targeted green prescription programs. However, there are limitations to the current research. Firstly, the use of an online survey could exclude groups who are less likely to have internet access or access to social media. Therefore, populations on lower incomes, with lower levels of computer literacy and older adults, may be under-represented in this study [39]. In addition, activity patterns were self-reported, future research using objective measurement of physical activity alongside diaries may provide a more accurate analysis of activity patterns, as well as the intensity of activities. In terms of the sample analysed, the entire population of anglers with disabilities were of white ethnicity, and the sample was predominantly older men. Future qualitative or mixed-methods research is needed to explore barriers and facilitators to fishing among women, younger males and under-represented ethnic groups to understand if our gender, ethnicity and age results were skewed as a result of sampling and recruitment bias, or as a result of barriers to fishing among these populations. In addition, there was no sub-group analysis of people with different types of disability (e.g., visually impaired and cognitive impairment), and no sub-group analysis of people with multiple physical or mental co-morbidities. Barriers and facilitators may be different across sub-groups, for example, people who are unable to drive due to their disability may be more likely to find a lack of public transport a barrier to fishing than people who can drive. However, the research did suggest that fishing may be more accessible than other sports in general for people with disabilities, which is important for addressing the overall inequity in sports participation between populations with a disability compared to those without a disability. It is also important to highlight that this research is not advocating for the prescription of fishing for all older men with disabilities, rather the research advocates for fishing to be one option among a range of options, which could improve the uptake of social green prescriptions for men with disabilities, in particular older men with disabilities.

## 5. Conclusions

Overall, fishing of all types has a higher engagement among men with disabilities, particularly among older age groups, relative to other sports. Such findings suggests that fishing could be a more feasible and acceptable ‘green prescription’ for older men with disabilities in comparison to other outdoor physical activity. Therefore, referral pathways which allow fishing to be included in the remit of outdoor activities which patients can be referred to, should be co-developed with people who have disabilities, to improve targeting of currently underserved populations. The challenge of fishing and the relaxation benefits should be emphasised to encourage greater uptake of fishing. In addition, addressing barriers of cost, transport and having no one to go with could improve accessibility of fishing and address inequities in fishing accessibility.

## Figures and Tables

**Table 1 ijerph-19-04730-t001:** Descriptive characteristics.

	No Disability (*n* = 1499)	Disability (*n* = 292)
**Age**		
18–24 years	2.9% (43)	1.4% (4)
25–34 years	9.5% (143)	6.8% (20)
35–44 years	18.6% (279)	15.1% (44)
45–54 years	21.1% (316)	20.2% (59)
55–64 years	24.7% (371)	29.5% (86)
65–74 years	19.7% (296)	21.2% (62)
75+ years	3.4% (51)	5.8% (17)
**Gender**		
Male	97.8% (1466)	96.2% (281)
Female	2% (30)	3.1% (9)
Non-binary	0.1% (2)	0.3% (1)
**Location**		
England	93.6% (1403)	92.8% (271)
Wales	2.3% (35)	5.1% (15)
Scotland	1.9% (29)	1.4% (4)
Northern Ireland	1.7% (25)	0.7% (2)
**Ethnicity**		
White	99% (1485)	100% (292)
Black	0.2% (3)	0 (0)
Asian	0.1% (1)	0 (0)
Mixed/multiple	0.6% (8)	0 (0)
Other	0.1% (2)	0 (0)
**Activity levels**		
Sitting time (h/day)	3 (2–3)	2 (2–3)
Screen time (h/day)	2 (1–4)	2 (1–4)
Time spent outdoors (h/day)	2 (2–2)	2 (2–2)
Time spent in moderate-vigorous physical activity (h/day)	2 (1–3)	2 (1–3)

Note: Data presented as percentage (*n*) for categorical variables and median (interquartile range) for continuous variables.

**Table 2 ijerph-19-04730-t002:** Fishing behaviours.

	Disability	No Disability	*p* Value
**Type of fishing**			
Coarse fishing	42.5% (124)	40.1% (601)	0.17
Sea fishing	2.4% (7)	2.7% (40)	
Match fishing	7.5% (22)	4.5% (68)	
Fly fishing	4.1% (12)	5.4% (81)	
Carp fishing	34.9% (102)	35.8% (536)	
Predator fishing	3.8% (11)	5.5% (82)	
Specialist fishing	3.8% (11)	5.6% (84)	
Other	1% (3)	0.4% (6)	
**Frequency of fishing**			
Everyday	0% (0)	0.1% (1)	0.13
5–6 times per week	0.7% (2)	0.8% (12)	
3–4 times per week	7.9% (23)	6% (84)	
1–2 times per week	39.4% (115)	37.9% (561)	
Once every two weeks	18.5% (54)	25.5% (401)	
Once every month	17.1% (50)	16.4% (242)	
Once every 2–3 months	11% (32)	9% (128)	
Once every 4–6 months	2.7% (8)	2.7% (40)	
Less than once every 6 months	2.7% (8)	1.8% (24)	
**Length of time spent fishing**			
Under one hour	0 (0%)	0.1% (2)	0.11
1–2 h	0.3% (1)	1.5% (22)	
3–4 h	16.4% (48)	12.4% (186)	
5 h or more	83.2% (243)	86% (1287)	
**Participation in fishing matches**			
Yes	20.6% (60)	18.4% (275)	0.38
No	79.4% (231)	81.6% (1218)	

Note: Data presented as percentage (*n*) for categorical variables.

**Table 3 ijerph-19-04730-t003:** Motivators to fishing.

	Disability	No Disability	*p* Value
**Motivators**			
To socialise	4.1% (12)	3.4% (50)	0.09
Enjoy the challenge of fishing	41.6% (121)	48.4% (718)	
To be outside	18.2% (53)	16.7% (247)	
To relax	32.6% (95)	30.1% (446)	
To catch food to eat	0% (0)	0.1% (1)	
Other	3.4% (10)	1.4% (21)	

Note: Data presented as percentage (*n*) for categorical variables.

**Table 4 ijerph-19-04730-t004:** Barriers to fishing.

	Disability	No Disability	*p* Value
Weather	53.1% (155)	51.3% (769)	0.56
Cost	24% (70)	10.1% (152)	**<0.001**
Lack of transport	19.2% (56)	9.7% (146)	**<0.001**
Lack of fishing venues nearby	11% (32)	10.7% (161)	0.91
No one to go with	17.1% (50)	6.7% (101)	**<0.001**
Time restrictions	24.3% (71)	52.4% (785)	**<0.001**
Other	21.2% (62)	10.3% (1334)	**<0.001**

Note: Data presented as percentage (*n*) for categorical variables.

## Data Availability

The data that support the findings of this study are available from the corresponding author, [R.K.L] upon reasonable request.

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
