# Peer review of "Fishing Participation, Motivators and Barriers among UK Anglers with Disabilities: Opportunities and Implications for Green Social Prescribing"

_ijerph, 2022, doi:10.3390/ijerph19084730_

Round 1

Reviewer 1 Report

Dear Authors,

Thank you for inviting me to review your manuscript, which I have just finished reading. Please find my notes below: 

  • The introduction provides ample background and includes a number of relevant references. 
  • The research design is appropriate for this type of study.
  • The results are more than adequately reported and discussed. 

My only question mark about this piece of research is as follows. It seems popular belief that fishing an activity mostly older white males are interested in. I wonder if there is previously published peer-reviewed evidence that can corroborate that fact? If this had been done at the time of designing the survey, you could have included some questions that allowed both white and non-white participants, including females of all ages as well as younger males, to express their views about fishing. I am aware that you state this should be the subject of future research, but seeing that Angling Direct has partly funded the study,  you may wish to do that you may want to reflect on whether it would be appropriate to be more explicit about how your failure to take popular belief into consideration ended up posing a limitation to the study. This may deter more sceptical reviewers and readers from seeing future research as an opportunity for your funding party to get involved in further research. Other than that, I congratulate you on an excellent piece of research. 

All best wishes,

The reviewer.

Author Response

We would like to thank reviewer 1 for their positive comments and valuable suggestions. We have now developed our limitations section to be more explicit regarding the limitation of our sampling procedure. The limitation section now includes: ‘in terms of the sample analysed, the entire population of anglers with disabilities were of white ethnicity, and the sample was predominantly older men. Future qualitative or mixed-methods research is needed to explore barriers and facilitators to fishing among women, younger males, and under-represented ethnic groups to understand if our gender, ethnicity and age results were skewed as a result of sampling and recruitment bias, or as a result of barriers to fishing among these populations.’

Reviewer 2 Report

It is a very interesting, very well conducted work – very well written. With questions coming up when reading e.g. around different types of disabilities, it is very well explained in the discussion section, where further investigation would be needed. I recommend publishing the paper after two very small additions.

It would be nice to have the information somewhere, which social media channels have been used and the name of the angler´s website (and visitor numbers of the website, if this data is available). Perhaps it is already included and part of the supplementary material - the link to the supplementary material did not work.

As every approach collecting information has some forms of limitation, it would be good to have a short sentence somewhere reflecting that the chosen online approach might collect less feedback from groups with less or a lack of digital skills or low affinity for using websites or social media. Therefore, some groups (e.g. elder persons) might be less represented in the sample.

Author Response

Thankyou to reviewer 2 for the consideration of our manuscript and their supportive comments. We have now modified our methods section to state our use of social media channels and websites in the distribution of the survey, unfortunately the visitor numbers of the website were unavailable. We have included the following text to the methods section: ‘ via an online survey that was advertised through Angling Direct and Tackling Minds Instagram, Facebook and Twitter accounts. Angling Direct also sent the survey link to their mailing list and the link was distributed via the Anglia Ruskin University Twitter account.’

We believe the link to the supplementary material will be updated by MDPI to allow readers to access the supplementary material uploaded with the manuscript. 

In regards to the second point raised by reviewer 2, we have now highlighted the limitations of online surveys by including the following text in the discussion: 'firstly, the use of an online survey could exclude groups who are less likely to have internet access or access to social media. Therefore, populations on lower incomes, with lower levels of computer literacy, and older adults, may be under-represented in this study [39].'